# Effect of Agave Fructans on Changes in Chemistry, Morphology and Composition in the Biomass Growth of Milk Kefir Grains

**DOI:** 10.3390/microorganisms11061570

**Published:** 2023-06-13

**Authors:** Germán R. Moreno-León, Sandra V. Avila-Reyes, Julieta C. Villalobos-Espinosa, Brenda H. Camacho-Díaz, Daniel Tapia-Maruri, Antonio R. Jiménez-Aparicio, Martha L. Arenas-Ocampo, Javier Solorza-Feria

**Affiliations:** 1CEPROBI-Instituto Politécnico Nacional, Carretera Yautepec-Jojutla, Km. 6 calle CEPROBI No. 8, Colonia San Isidro, Yautepec C.P. 62730, Morelos, Mexico; gmorenol2100@alumno.ipn.mx (G.R.M.-L.); bcamacho@ipn.mx (B.H.C.-D.); dmaruri@ipn.mx (D.T.-M.); arjaparicio@gmail.com (A.R.J.-A.); arenasocampoml@gmail.com (M.L.A.-O.); 2CONAHCyT- CEPROBI-Instituto Politécnico Nacional, Carretera Yautepec-Jojutla, Km. 6 calle CEPROBI No. 8, Colonia San Isidro, Yautepec C.P. 62730, Morelos, Mexico; 3Tecnológico Nacional de México/Campus ITS Teziutlán, Ingeniería en Industrias Alimentarias, Fracción I y II Aire Libre S/N, Teziutlán C.P. 73960, Puebla, Mexico; julieta.villalobose@gmail.com

**Keywords:** agavins, lactose-free milk, microscopy, *Agave angustifolia*, image analysis, response surface

## Abstract

Prebiotic effects have been attributed to agave fructans through bacterial and yeast fermentations, but there are few reports on their use as raw materials of a carbon source. Kefir milk is a fermented drink with lactic acid bacteria and yeast that coexist in a symbiotic association. During fermentation, these microorganisms mainly consume lactose and produce a polymeric matrix called kefiran, which is an exopolysaccharide composed mainly of water-soluble glucogalactan, suitable for the development of bio-degradable films. Using the biomass of microorganisms and proteins together can be a sustainable and innovative source of biopolymers. In this investigation, the effects of lactose-free milk as a culture medium and the addition of other carbon sources (dextrose, fructose, galactose, lactose, inulin and fructans) in concentrations of 2, 4 and 6% *w*/*w*, coupled with initial parameters such as temperature (20, 25 and 30 °C), % of starter inoculum (2, 5 and 10% *w*/*w*) was evaluated. The method of response surface analysis was performed to determine the optimum biomass production conditions at the start of the experiment. The response surface method showed that a 2% inoculum and a temperature of 25 °C were the best parameters for fermentation. The addition of 6% *w*/*w* agave fructans in the culture medium favored the growth of biomass (75.94%) with respect to the lactose-free culture medium. An increase in fat (3.76%), ash (5.57%) and protein (7.12%) content was observed when adding agave fructans. There was an important change in the diversity of microorganisms with an absence of lactose. These compounds have the potential to be used as a carbon source in a medium culture to increase kefir granule biomass. There was an important change in the diversity of microorganisms with an absence of lactose, where the applied image digital analysis led to the identification of the morphological changes in the kefir granules through modification of the profile of such microorganisms.

## 1. Introduction

Milk Kefir is a beverage that is consumed for its beneficial health effects due to the bioactive compounds produced by milk kefir grains during fermentation [1,2]. Kefir beverages can be classified according to the medium in which the bacteria are cultivated during fermentation, which can be sugar water or milk through lactose fermentation. Due to the above, the kefir grains have different physical and chemical characteristics, as well as a different microbiological composition [2,3]. Kefir grains have a small cauliflower shape, with dimensions ranging from 10 to 30 mm, displaying a white to yellowish color, with lobules, firm texture and irregular shapes [4]. They are made up of a symbiotic mixture of organized groups of microorganisms (lactic acid bacteria, acetic acid bacteria and yeasts), which are included in a polymeric matrix of exopolysaccharide called kefiran [5]. Bacteria and yeast interact with each other during the fermentation process. For example, yeasts provide vitamins, amino acids, simple sugars and growth factors for bacteria, and these, in turn, use it for their metabolic process. On the other hand, Kefiran is a extracellular polysaccharide, described as a water-soluble glucogalactan, produced by microorganisms present in kefir grains, such as: *Lactobacillus kefiranofaciens*, *L. kefir*, *L. kefirgranum* and *L. parakefir*, among others [6,7,8]. This exopolysaccharide has been used in the food, pharmaceutical and cosmetic industries as additives, bio-absorbents, bio-flocculants and as a drug release agent [9,10]. In the last decade, it has been reported that this type of biopolymer improves the viscosity and viscoelastic properties of dairy product gels [11]. It also has the ability to form edible, flexible and homogeneous films with good mechanical and barrier properties [12]. However, the production costs to obtain exopolysaccharides are still high, mainly due to carbon sources (sugars) required in the growth media [13].

The composition of the culture media (source of carbon, nitrogen and vitamins and minerals) and the temperature affect the production and physico-chemical composition of kefiran and the microbiota of the kefir grain, so it is important to optimize the culture media to achieve an increase in the kefir biomass and use it in the mentioned commercial applications [14,15]. The use of alternative sources of carbon such as agroindustry products is an option that could help reduce the polluting residues that they generate while providing added value [16]. Polymers, such as starch, cellulose and fructans, contain glucose or fructose monosaccharides in their composition, a carbon source that is mainly metabolized by yeasts [17].

In recent years, the prebiotic effect and functional efficiency of fructans have been reported in more than one species of plants, due to the fact that they are fermentable by microorganisms, such as bacteria and yeasts [18,19]. Fructans are the most abundant polysaccharides in nature, representing approximately 15% of flowering plants. They are mixtures of fructose oligomers and polymers linked by β (2→1) and/or β-(2→6) bonds, synthesized from a sucrose molecule [20]. The type of bond between the adjacent fructose monomers, the position of the glucose residues and the degree of polymerization determine the types of fructans, which can have, in their conformation, from three to hundreds of fructose units [21]. In Mexico, 159 species of the agave genus have been reported, 116 (75%) are located in Mexico and 90 of these species (58%) are endemic. *Agave angustifolia* Haw is an endemic plant of México; the stem or “pineapple” is the most important part of the plant because the concentrates of fructans and fermentable sugars are used to make alcoholic beverages, such as mezcal and bacanora [22,23]. In this context, the use of agave fructans is an alternative carbon source for fermentation cultures and to produce kefir biomass. It has been reported that bacteria, such as *L. paracasei*, *L. acidophilus*, *L. rhamnosus*, *L. reuteri* and yeasts of the genus *Saccharomyces*, can metabolize fructans of different degrees of polymerization [21,24]. The revision of different carbon sources and culture media alternatives without a lactose source can be a strategy to increase the biomass of kefir grains or, otherwise, obtain alternative materials for fermentation [25].

Image digital analysis (IDA) is an auxiliary tool for the analysis of texture images, which has been used to quantify changes in biological samples such as kefir grains because the texture of the image reflects changes in intensity pixel values, yielding information about the color and geometric structure of objects [26,27]. The texture image analysis represents the spatial disposition of the gray levels in pixels of a region, obtaining characteristics for the description of changes in the areas of objects by roughness and homogeneity [28]. Likewise, the Grayscale Co-occurrence Matrix (GLCM) and Shifting Different Box Counting (SDBC) methods are used to analyze texture changes.

Therefore, this work aimed to optimize fermentation conditions in lactose-free milk and later to evaluate the effect of different carbon sources to increase the biomass of kefir grains to replace reduced-fat whole milk with lactose-free milk to evaluate the physicochemical and morphological changes of kefir grains through image analysis.

## 2. Materials and Methods

### 2.1. Material

To establish the fermentation conditions, as culture media, whole milk (with lactose) and non-lactose milk of commercial origin, both with low fat content (0.6–1.13%), were used, both set as control groups and identified as low-fat whole milk (SM) (Grupo Lala, S.A.B. de C.V., Durango, México) and lactose-free milk (LFM) (Svelty, Nestlé México S.A de C.V.). After establishing the conditions for optimizing the fermentation process, a second fermentation was carried out using additional sources of carbon to the lactose-free milk.

As additional carbon sources, there were added monosaccharides: anhydrous dextrose (27740, Golden Bell, Querétaro, Mexico), fructose (USP, 08431, Fermont, Monterrey, Mexico) and galactose (G0750, Sigma-Aldrich, St. Louis, MO, USA); the disaccharide lactose (1.07657.1000, Millipore, Burlington, MA, USA); and fructooligosaccharides: chicory inulin (Beneo^TM^ ST, Orafti, Mexico) and agave fructans, from *Agave angustifolia* Haw., obtained at CEPROBI-IPN (patent 380041). As biological material, milk kefir grains from Teziutlán, Puebla province, were utilized.

### 2.2. Culture Conditions of Kefir Grain

For the maintenance of the kefir grains, a proportion of 10 g of kefir grains per liter of whole milk was used at room temperature for 30 days. The milk was changed every 24 h, considering the initial proportion of the starter culture. For their conservation, they were kept at −20 °C and reactivated according to the fermentation medium. For the growth, 2 g of kefir grains was inoculated into 100 mL of both types of milk (SM and LFM, 10% *w*/*w* total solids) following the methodology of Piermaria et al. (2008) [29] with modifications for their harvest. Both sets were used as control groups. The fermentations were carried out in a discontinuous system (in batch) or closed system in an incubator in static mode for temperature monitoring (LAB-LINE, Model R3525, Melrose, IL, USA). The incubation temperature was 30 °C, without agitation, and batch fermentation was carried out under ambient atmospheric conditions in all cases, up to 24 h. At each change in the culture medium, the kefir grains were filtered and washed with sterile water in a plastic sieve that was previously sanitized with 70% (*v*/*v*) ethylic alcohol [11].

### 2.3. Determination of Biomass of Kefir Grains

The optimum fermentation conditions for the biomass increase were determined using the modified methodology of Bezerra et al. (2019) [30], taking into account SM and LFM as culture media in the initial fermentation. For the experimental design, a factorial 2^3^ (two factors or variables at three levels: namely temperature and % inoculum) with three repetitions was used. The independent variables evaluated were the fermentation temperature *X*1 (20, 25 and 30 °C) and the initial inoculum concentration *X*2 (2, 5 and 10% *w*/*w*). The choice of optimization was based on the response variable *Y*1: biomass increase (% *w*/*w*).

The determination of the biomass increase in kefir grains was carried out via gravimetry [14,31]. The culture of the kefir grains was carried out according to Section 2.2. At each culture medium change, the kefir grains were carefully dried and weighed on an analytical balance (OHAUS model No. AR2140) to record the biomass increase by applying Equation (1). Successive cultures were carried out, and the biomass increase was recorded every 24 h for 7 days [32].
(1)$$Biomass\;increase\left(\%\right)=\left(\frac{M_{2}-M_{1}}{M_{1}}\right)\times \left(100\right)$$
where *M*_1_ = initial mass of the kefir grains and *M*_2_ = mass of the kefir grains after 24 h of fermentation. All experiments were undertaken in triplicate.

#### Optimization through Response Surface Methodology

A response surface analysis was performed to find the optimum conditions of the relationship between the independent variables (fermentation temperature and % inoculum concentration) and the response variable (% biomass increase) [33]. The mathematical model of the response was initially obtained via a first-order linear model, and the ANOVA analysis was applied to decide its significance degree (Equation (2)):(2)$$y=\beta_{0}{+\beta }_{1}X_{1}{+\beta }_{2}X_{2}+....\beta_{k}X_{k}+\varepsilon$$
where *y* denotes the approximated function of a set of design variables *X*_1_, *X*_2_, *X*_k,_ and the function can be approximated in some region of the xs using a polynomial model. The regression coefficients *β*_0_, *β*_1_, *β*_2_ and *β*_k,_ are parameters to be estimated from the data. The term ε is the experimental error, i.e., a measurement error on the response, plus the effect of other variables, and any other source of error.

The statistical analysis indicated that there was a curvature degree in the system, indicating that a first-order linear model was not appropriate and so it was reevaluated with a model with a higher order. The second-order model included all the terms in the first-order model, plus all quadratic terms such as $\beta_{ij}X_{i}^{2}$ and all cross-product terms or interaction as $\beta_{ij}X_{i}X_{j}$. As usual, this is expressed with a second-order quadratic model as follows (Equation (3)):(3)$$y=\beta_{0}+\sum_{i=1}^{k}\beta_{i}X_{i}+\sum_{i=1}^{k}\beta_{ii}X_{i}^{2}+\sum_{i=1}^{k-1}\sum_{i=1}^{k}\beta_{ij}X_{i}X_{j}+\varepsilon$$

The fitness of the model was verified through statistical analysis (determination coefficient of the regression, R^2^) with a significance level of *p* < 0.05.

### 2.4. Effect of Different Carbon Sources

Once the results were obtained from the response surface methodology, the best condition for a biomass increase was chosen in accordance with Section 2.1 and Section 2.3, as previously mentioned. To evaluate the effect of different sources of carbon added at the LFM, the following carbohydrates were added to the culture media: monosaccharides (dextrose (D), fructose (F) and galactose (G)), disaccharides (lactose (L)) and fructooligosaccharide (inulin (I) from chicory and fructans (LFM-F) from *A. angustifolia* Haw.) at concentrations of 2, 4 and 6% (*w*/*w*) of total solids.

### 2.5. Chemical Analysis

The chemical composition of the optimized biomass was determined (AOAC 1980) [34]. Moisture content was evaluated at 102 ± 1 °C (16.192), protein content using the Kjeldahl method (16.193), determination of total ash (16.196), fat content through Soxhlet extraction (13.031) and total carbohydrate content, using the phenolsulfuric acid method [35].

### 2.6. Microstructure Analysis

#### 2.6.1. Environmental Scanning Electron Microscopy

The microstructure of kefir grains treated with low-fat whole milk (SM), lactose-free milk (LFM) and lactose-free milk fortified with fructans (LFM-F) was observed using Environmental Scanning Electron Microscopy (ESEM) (Carl Zeiss, model EVO LS10, Jena, Germany). The grains were placed in a mortar, and liquid nitrogen was added for freezing. Then, they were placed on a conductive carbon tape and placed on a freezing plate. Micrographs of the surface and cross-section (lateral) of the grains were obtained at a gauge water vapor pressure of 30 Pa, with an accelerating voltage of 15 kV. All micrographs were stored in TIFF format with a resolution of 1024 × 768 pixels.

#### 2.6.2. Image Digital Analysis of ESEM

The micrographs captured with the ESEM were processed using the ImageJ v.1.50d software (National Institute of Health, Bethesda, MD, USA), and the fractal analysis was conducted using the box-counting method with the FracLac Box-Counting plugin. The parameters of texture (contrast, inverse difference moment, entropy, etc.) were analyzed using the Gray-Level Co-Occurrence Matrix (GLCM).

### 2.7. Gram Staining

Gram staining was performed to follow changes in the populations of the bacteria according to the changes in the culture media. Immediately after staining, the sample was mounted on a Canada Balsam Synthetic Mounting Medium. Observations were made using an optical microscope (Nikon Eclipse 80i, Tokyo, Japan). Images at 20× and 100× were obtained using a coupled digital camera (CCD MTI DC330, Oxford Ins., Abingdon, UK). The results were qualitatively evaluated.

### 2.8. Statistical Analysis

All analyses were conducted in triplicate, and statistical analysis was conducted using the Minitab 14.0 statistical software (Minitab Inc., State College, PA, USA) to perform one-way analysis of variance (ANOVA) with a significance level of 95% (*p* < 0.05). Significant differences among means were compared using post hoc of Tukey methodology (*p* < 0.05) when the statistical test was significant. Regression analysis with ANOVA was applied to a second-order quadratic model as well. Additionally, response surface analysis among variables was performed.

## 3. Results and Discussion

### 3.1. Kefir Grain Conditioning

Plots of the behavior of the increase in biomass of kefir grains are shown in Figure 1. These representations show the behavior of the fermentation kinetics for 7 days in two types of control milk (with lactose and lactose-free) as a culture medium. SM and LFM were used in both culture media with three concentrations of inoculum (2, 5 and 10% *w*/*w*) and three temperatures (20, 25 and 30 °C) to evaluate the effect of the initial inoculum concentration and the temperature with respect to biomass growth. The effect of temperature during fermentation on biomass production shows that on the second fermentation day, the % of biomass obtained was constant, like an adaptation phase. However, after day three, the 2% inoculated condition showed an increase in biomass production of almost 200% higher in LFM than in SM. Pop, Apostu and Salanţă, (2014) [32] reported the best optimum culture conditions at 25 °C for 24 h in a supplemented culture medium with lactose, glucose, galactose and sucrose, where they obtained a yield of about 38.9 g/L of kefir grains biomass. This fact was attributed to the time necessary to assimilate the carbon source, and if there was a positive assimilation, there was an increase in biomass. This has been reported due to the role played by bacteria and yeasts [8]. Other researchers found that the increases in biomass of kefir grains were mainly due to the grain formation capacity and the exopolysaccharide production. It has been reported that the cell surface properties of the kefir microbiota, together with the fermentation conditions, play an important role in their formation. For example, a hydrophilic surface is associated with the presence of polysaccharides, while on the contrary, hydrophobicity is an important factor that determines whether microorganisms adhere to the grain surface. Therefore, hydrophobicity could prevent microbial attachment to the grain surface, thus affecting the increase in kefir biomass [36,37].

However, the kefir grain usually contains between 83 and 90% of lactic acid bacteria of the mesophilic type and yeasts that are characterized by fermenting substrates at temperatures between 25 and 30 °C. The presence of these microorganisms and changes in temperature have an influence on the enzyme activity, as well as in the fermentation rate and the consumption of sugars present in the culture medium [38]. An increase in the concentration of the inoculum and the solids present in the culture medium will reduce the rate of absorption of the carbon source in the medium by each cell, presenting stress due to population density and competition for nutrients. This could mean a decrease in the biomass of the kefir grain or delay in the fermentation process [39].

### 3.2. Analysis of Biomass Increase in Conditioning

The LFM culture medium showed significant values for biomass optimization, showing values of 19 ± 6.93% up to a 20-fold increase (460.62 ± 8.12%), modifying the temperature conditions and % of initial inoculum. On the other hand, for the treatments, regardless of the type of milk and temperature used, the values that showed the greatest increase were those used at a concentration of 2% of inoculum. Likewise, the fermentations with SM showed that despite having values of biomass up to 248.3 ± 27.2% at the highest concentration of inoculum, they were not compared with that obtained for the treatment with LFM, since the biomass production resulted in a double production, maintaining the conditions mentioned above.

Regarding the temperature, the one that showed the greatest increase in biomass was at 25 °C, regardless of the initial inoculum (2, 5 or 10%). The results showed that 2% of the initial inoculum with LFM (460.62 ± 8.12%) had a better response compared to SM (248.3 ± 27.2%), at the same conditions, during five days of the fermentation. The increase in the usual percentage of the inoculum (5% and 10%) affected the increase in biomass. Apar et al. (2017) [14] reported that a traditional inoculation had a relatively low performance value when it was used in a commercial application. The results obtained showed that the growth rate was inversely proportional to the amount of the initial inoculum, obtaining a greater biomass growth (2% inoculum) and a lower growth of 10% inoculum for all temperatures. Some authors reported that a temperature range between 25 and 27 °C is best to carry out the fermentation of kefir [1,14,31] since, at higher temperatures, there was a loss of grains due to the effect of temperature, as observed in this work for both types of culture medium: SM with lactose and lactose-free milk. This temperature range (25–27 °C) was where the best biomass growth was observed, unlike 20 °C and 30 °C, regardless of the inoculum concentration.

### 3.3. Kefir Grains Conditioning Optimization

The analysis of the response surface is shown in Figure 2A,B. When observing both variables (temperature and inoculum concentration), there was no overall notorious correlation between the dependent variable, the % biomass with respect to temperature and the percentage of inoculum, which suggests that the observed trend was particularly attributed to the type of milk used. Based on the response surface analysis, Figure 2A shows an ANOVA applied with regression models for the different types of milk. For the treatment with the SM, the biomass relationship with respect to the independent variables was adjusted to a linear quadratic model, obtaining values of R^2^ and R^2^ adjusted to 0.9082 and 0.7252, respectively, presenting the *X*2 factor or the inoculum concentration as one of the independent variables with interaction with respect to the biomass increase. There was a significant difference (*p* < 0.05) with respect to the rest of the source of variation and its combinations.

The applied model explained more than 70% the results obtained (R^2^ > 0.7). The graph in Figure 2A of the response surface for the SM culture medium indicates that the percentage of biomass was favored by decreasing the initial concentration of the inoculum; however, even though there was no uniform trend when increasing or decreasing the temperature, the statistical analysis showed that the optimum condition was a temperature value of 30 °C with an inoculum concentration of 2%.

The LFM culture medium (Figure 2B), such as the SM culture medium, had a direct and significant interaction (*p* < 0.05) with respect to the inoculum concentration value, having, for this, an adjustment of 0.8678 and 0.6476 for R^2^ and R^2^ adjusted, respectively, and being significant with the applied model explaining more than 60% of the results obtained. Overall, for both models according to the medium used (SM and LFM), the interactions between factors *X*_1_
*X*_2_ (Equation (3)) did not present significant interaction. When using LFM, both independent variables (temperature and inoculum) modified the behavior of the response variable (Figure 2B), in turn, for the case of the SM; as the inoculum concentration increased, the biomass concentration decreased proportionally (Figure 2A). However, for the LFM culture medium, the increase with the same percentage of inoculum (2%) doubled the percentage of biomass. In addition, the temperature used for this culture medium indicated that the behavior of biomass production was not favored proportionally, since the intermediate value of 25 °C turned out to be optimal, compared to limited values between 20 and 30 °C, which did not favor an increase in the response variable.

The response surface models were adjusted to obtain the probability values in an equation based on the mentioned linear-quadratic model to finally obtain the regression Equations (4) and (5), as well as optimal growth conditions based on a confidence interval with a *p* < 0.05. Thus, in the case of SM (Figure 2A), the optimal conditions to produce about 246 ± 3.6% biomass were a temperature of 30 °C and 2% inoculum, while the optimization of the process for LFM showed that the best biomass concentration was obtained at a temperature of 25 °C and 2% of inoculum, obtaining a higher biomass value than for SM of up to 460.62 ± 8.12% (Figure 2B).

Several authors have shown that the carbon source concentration, temperature and the amount of inoculum are significant factors in the biomass production, which is consistent with the response surface analysis presented, since, in this case, the temperature and % inoculum significantly affected the biomass production of kefir grains [40]. In addition, these same effects of the factors were evaluated for the increase in kefiran, and it has been previously reported that this increase in biomass of kefir granules has a fermentation temperature range between 20 and 30 °C, with a maximum production of kefiran, as well as an optimal pH value, as reported between values of 5 and 6 [9,31,41]. Moreover, the individual effects and interaction of carbon source, pH and temperature for the increase in biomass of kefir granules were reported and evaluated by means of a polynomial equation of second order, showing that R^2^ and R^2^ adjusted had a goodness of fit for the analysis of the model of 0.898 and 0.851, respectively. For the proposed model, precision (F value) of 18.78 was found, which indicated that it was adequate, concluding that the percentage weight of the granule increased when pH and the percentage of solids in the milk increase but the growth rate was reduced when the solids of milk concentration were too high [31].

Regression equations in coded units:% Biomass LFM (g/L) = −3335 − 104.8A + 304B + 5.05A^2^ − 5.97B^2^ + 0.63AB(4)
% Biomass SM (g/L) = −500 + 5.6A + 46.5B + 1.60A^2^ − 0.610 B^2^ − 1.639AB(5)

### 3.4. Effect of Carbon Source and Culture Medium

Following the response surface analysis (Section 3.3), the best growth condition was used to evaluate the behavior of the kefir grains, enriching the culture medium with different carbohydrates (dextrose, fructose, agave fructans, galactose, inulin and lactose). Figure 3A shows the growth of biomass with a concentration of carbohydrates added to the culture medium at 2, 4 and 6%. The milk control at this stage was only lactose-free milk as a culture medium, as there were significant differences with some of the enriched growth media. The analysis of means (ANOVA) resulted in at least three statistically significant groups that were favored by the addition of carbohydrates and the changes in concentrations. The culture medium enriched with agave fructans was the one that most favored the growth of biomass (75.94%) with respect to the control, where the concentration of fructans (4 and 6%) and fructose (6%) as carbon sources significantly affected the biomass increase. With the change of milk along with the culture medium, a change in the morphology of the grains was observed, decreasing the size from 15 mm to 2 mm of thickness (Figure 3B), but an increase in the volume and weight of the biomass was obtained, with different characteristics from those of a traditional grain. The biomass showed a smooth and compact texture, completely losing the cauliflower shape of the grains. One hypothesis for the cause of this increase is that there was a change in the population density of microorganisms, with the absence of lactose as the main sugar for lactic acid bacteria, holding the prevalence of other microorganisms such as some yeasts. The yeasts have the ability to ferment carbon sources, such as oligosaccharides, sucrose, maltose, lactose, galactose, glucose and linear fructans (inulin) and branched fructans (agave fructans) [1,42,43].

However, by adding agave fructans, the protein content increases (7.12%), as well as the ash and fat value (5.57%) (Section 3.5). Therefore, it can be inferred that the increase in kefir biomass could be due to an encapsulation of coagulated casein micelles in the exopolysaccharide produced by some of the bacteria that are homofermenters of hexoses, such as glucose and fructose. In a previous study carried out by our working group, the use of whole milk with lactose enriched with carbohydrates, such as glucose, fructose, galactose and fructans, was reported, reaching up to 220% of kefir biomass using agave fructans and 190% with galactose, preserving, at all times, the classic cauliflower morphology of the kefir grains [43]. It has been reported that this behavior is due to the fact that the optimization in the production of kefir biomass is given by the carbon source for cell growth and metabolism of these sugars, mainly due to the capacity of bacteria and yeasts that are present in the kefir grain [1].

### 3.5. Chemical Analysis

The chemical composition of the kefir grain fermented in SM as a control, LFM and LFM-F is shown in Table 1. The results presented significant differences (*p* < 0.05), showing that the highest moisture content was for SM (82.5 ± 0.92%), and for carbohydrates, it was for LFM (11.92 ± 0.72%); however, the content of fat (3.76 ± 0.42%), ash (5.57 ± 0.85%) and protein (7.12 ± 0.49%) was much higher for the biomass of kefir grains grown in LFM-F. A higher solids content increased the osmotic stress, which caused a decrease in moisture content; however, it has been reported that this stress also promotes the increase in fats due to the presence of glycerol in the medium because it has the function of being osmoprotective for cells. It has been reported that some yeasts have lipolytic activity under these conditions, as are some species of Saccharomyces [44,45], which supposes an increase in this type of microorganism proportionally to the increase in sugars in the medium. In addition, lactose-free milk had an increase in the amount of glucose and galactose and, in turn, the enrichment of this same medium with agave fructans. Schoevers and Britz, (2003) [46], working with kefir grains fermented in SM, reported 82.6–83.5% moisture content, 1.35–1.69% of fat content and 6.34–7.76% of carbohydrate content, which are similar to those obtained in this work. Other studies have shown that fermentation conditions affect the physical, chemical and functional characteristics as well as the production of exopolysaccharides to maintain the matrix of kefir grains. A previous work reported some culture media based on soy or coconut drinks with water kefir grains, showing some similarity to the consistency of milk, to develop vegan or lactose-intolerant kefir drinks [42]. However, so far, no reports have been found for fermentation with the use of milk kefir grains in lactose-free milk.

### 3.6. Microstructure Analysis

To visualize the changes in the microstructure of the kefir grains and the biomass exerted by the culture medium containing SM and the other with lactose-free milk LFM, ESEM images were obtained (Figure 4). In the kefir grain fermentation processes, the morphological characterization of the conglomerate of microorganisms (bacteria and yeast) is relevant since the carbohydrates used could modify their morphology, which is a factor that could influence the biomass production and, with this, the final characteristics of the products or subproducts obtained from the fermentation process. Complementarily to the description of the increased biomass, textural features based on the Gray-Level Co-Occurrence Matrix (GLCM) algorithm were taken from superficial and cross-sectional or lateral kefir grains from SM, LFM and LFM–F samples, and their values are listed in Table 2 (surface of kefir grains) and Table 3 (cross sections of kefir grains). Related to micrographs of SEM milk kefir grains, the correlation values did not show significant differences (*p* < 0.05). However, the other parameters of texture (contrast, inverse difference moment (IDM) and entropy) showed significant differences (*p* < 0.05).

Contrast is a measure of local variations in gray-level values of image pixels [47]. As mentioned above, the contrast and entropy showed significant differences (*p* < 0.05) between SM and LFM–F samples. This parameter let us to identify the homogeneity of the image surface, which relates to the presence of fructans as a carbon source in the fermentation process, which, in turn, affect the morphology of the milk kefir grain, as evidenced by high-heterogeneity morphologies (Figure 4). Avila-Reyes et al. (2022) [43] reported that the use of fructans as a carbon source generates more complex morphologies in milk kefir grains. In addition, IDM is a measure that represents the local homogeneity of the image where high values show high local homogeneity in an image [48].

It is worth noticing that the morphology of the samples was affected by the type of carbohydrate employed, where it could be observed that the samples conditioned with low-fat milk (SM) had 16% more local homogeneity than the LFM and LFM-F. This behavior could probably be related to the process of adaptation to the new environment of carbon source without lactose. Wang et al. (2018) [49] reported that the long- or short-shaped rods presented by LAB showed evidence of their adaptation on the carbon source employed. The heterogeneity presented by the samples was confirmed by the results obtained in the entropy values, where the sample LFM–F presented a high value (8.17). These results could be related to the fermentation kinetics, in which the kefir grains fermented in milk without lactose presented a lower increase in biomass. Therefore, for the fermentation conditions used, the high heterogeneity of the image indicated a lower increase in biomass.

To identify if the morphological change was reflected in the same way inside the kefir grain, a cross-sectional image was made and analyzed. The results showed that the absence of lactose modified the distribution of the microorganisms. The image analysis of LFM and LFM-F samples showed the presence of bacteria on the external surface cross-section (Table 3). This presence could not be perceived in the kefir grain fermented with low-fat milk (Figure 2B). The presence of bacterial cultures generated a morphology with the presence of cavities and, with it, a high heterogeneity in the internal microstructure of the kefir grains. Păcularu-Burada et al. (2022) [50] reported variations in the microbiota of milk or water kefir grain related to the composition of carbon source in the fermentation. In addition, Wang et al. (2018) reported that the SEM images provided evidence for trophic adaptation to the hollow globular grain structure of Tibetan kefir grains of *L. kefiranofaciens*, exhibiting two distinct morphotypes, a short rod (3.0 μm in length) and long rod (10.0 μm in length), upon the colonization of either the outer surface or the inner component of Tibetan kefir grains.

The fermentation conditions, such as culture medium, carbon source added or temperature, will not only affect the growth rate but also the symbiotic association of the microorganisms that make up the kefir grain and their ability to ferment the milk components [51], such as those of bacteria and yeasts [37,52]. To observe changes in the abundance and population types of microorganisms with the change in culture medium from SM to LFM and LFM fortified with fructans (LFM-F), Gram staining was performed (Figure 4). This staining is a method to classify bacterial species into two groups: Gram-positive and Gram-negative, according to the properties of their cell wall, highlighting the Gram positives for preserving blue coloration due to the peptidoglycan layer, unlike the Gram negatives that lose this coloration, preserving only a red or pink coloration, which also allows for observing morphology, size or aggregations of the bacteria [53]. The classification of microorganisms present in the kefir grain is based mainly on morphology and fermentation mode (homofermenting and heterofermenting). In Figure 4, in low-fat milk (SM), where lactose is still present, the predominant population is Gram-positive bacteria, along with mainly lactic acid bacteria (LAB). In previous studies, it has been reported that *Lactobacillus kefiranofaciens* and *Lactobacillus kefiri* are the main bacteria responsible for the formation of exopolysaccharides [13,54,55], in addition to others such as *Leuconostoc mesenteroides*. These microorganisms are mainly Gram-positive, homofermenters, unlike *Leuconostoc*, which is a heterofermenter bacterium, which all use lactose or glucose as their main carbon source [56].

In lactose-free milk, the lactose is hydrolyzed so that sugar is present as glucose and galactose. In Figure 5, LFM and LFM-F presented a change in the diversity of the microorganisms present, decreasing Gram positives and increasing Gram negatives, in addition to increasing the yeast population, as can also be seen in Figure 3, ESEM. The presence of yeasts is crucial for the desirable properties in products such as kefir. These microorganisms may compete for growth nutrients or produce metabolic products to stimulate the growth of others. Very few yeast isolates in kefir are lactose-positive, but most strains can utilize galactose, lactate or citrate [57]. Previous studies have reported that yeasts, such as *Kluyveromyces marxianus* and *Saccharomyces turicensis,* are responsible for the lactose degradation process by enzymes, such as β-Dgalactosidase and lactase, as well as fructanases [58], for the conditioning of the medium and for the self-aggregation of the kefir grain [59,60,61]. Fructanases are responsible for breaking β-2-1 or β-2-6 bonds in fructans. Some examples of this type are exo-β-fructosidases (exoinulinases and exolevanses) and β-2-6-fructan-6-levanobiohydrolases [62,63]. Therefore, the addition of carbohydrates such as agave fructans made up mainly of fructose chains linked to glucose possibly influences the growth of yeasts such as *Kluyveromyces marxianus.*

## 4. Conclusions

From this study, it was found that the optimization of the biomass production of kefir granules depended on temperature, type of fermentation medium and % of inoculum, because they significantly affected the growth of kefir granules; however, the factor that most influenced biomass production was the concentration of inoculum. Statistical analysis showed that a 2% inoculum of kefir granules was the most effective for the LFM and SM culture media. This study provided information about how the composition and origin of the carbon source used or added to the culture medium, such as agave fructans, had a direct effect on the conformation of the kefir grains, leading to its biomass optimization. It was observed that the ratio of fat and sugar of the milk used (whole or lactose-free) gave rise to changes in the population density of microorganisms and, therefore, in the decrease in the production of the exopolysaccharides, leading to kefir grain formation. When using lactose-free milk, the presence of caseinate micelle coagulates was obtained as part of the biomass, but it did not promote the typical formation of small cauliflowers clustered together in the kefir grain. The results showed changes in the physicochemical and morphological characteristics of the kefir when agave fructans were incorporated as a carbon source. Further research is needed, but, so far, this work has shown that fructans derived from agave have the potential to be used as a medium culture to produce kefir granule biomass, reducing the use of whole milk with lactose as the main carbon source, considering that milk is a staple food, which, somehow, has limited the use of kefir worldwide.

## Figures and Tables

**Figure 1 microorganisms-11-01570-f001:**
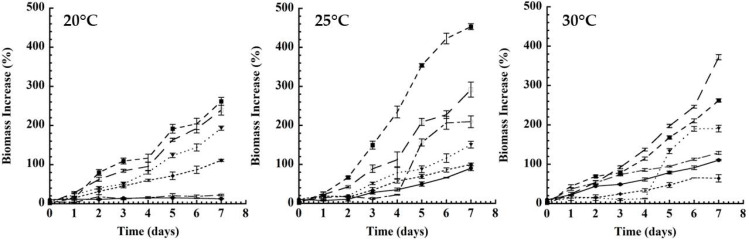
Kefir fermentation kinetics at different temperatures and initial inoculum. Whole low-fat milk (SM) and lactose-free powdered milk (LFM), where: SM 10% (--♦--); SM 5% (- -
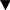
- -); SM 2% (-□―); LFM 10% (―●―); LFM 5% (-o―); LFM 2% (-

–).

**Figure 2 microorganisms-11-01570-f002:**
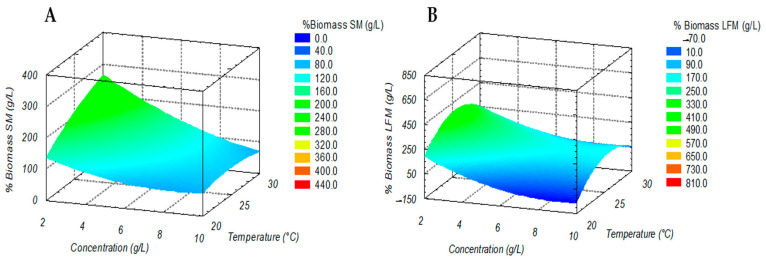
Optimization from starter cultures, where: (**A**), SM: low-fat whole milk and (**B**), LFM: lactose-free milk.

**Figure 3 microorganisms-11-01570-f003:**
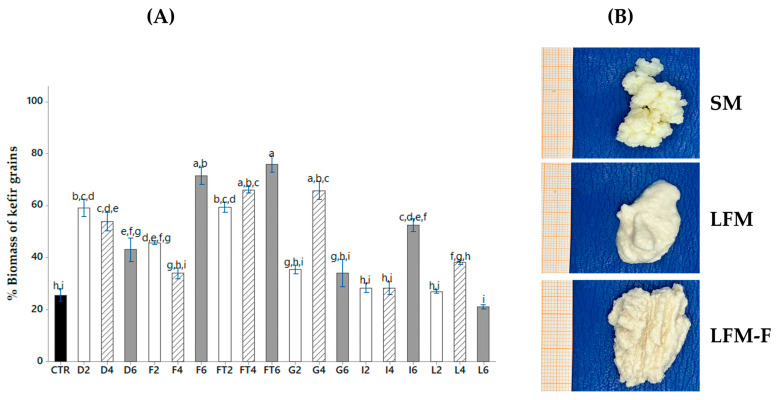
Effect of the different carbohydrates on the % biomass of the kefir grains obtained over 24 h. (**A**) Growth of biomass, CTRL: Control (

), D: dextrose; F: fructose; FT: fructans; G: galactose; I: inulin; L: lactose, in a proportion of 2 (
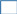
), 4 (
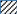
) and 6 (
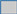
) % (*w*/*w*), respectively. Results are expressed as mean ± SD; (*p* ˂ 0.05), Different letters indicates statistically significant differences using an ANOVA followed by Tukey test. (**B**) Kefir grains; SM: low-fat whole milk; LFM: lactose-free milk; LFM-F: lactose-free milk with fructans.

**Figure 4 microorganisms-11-01570-f004:**
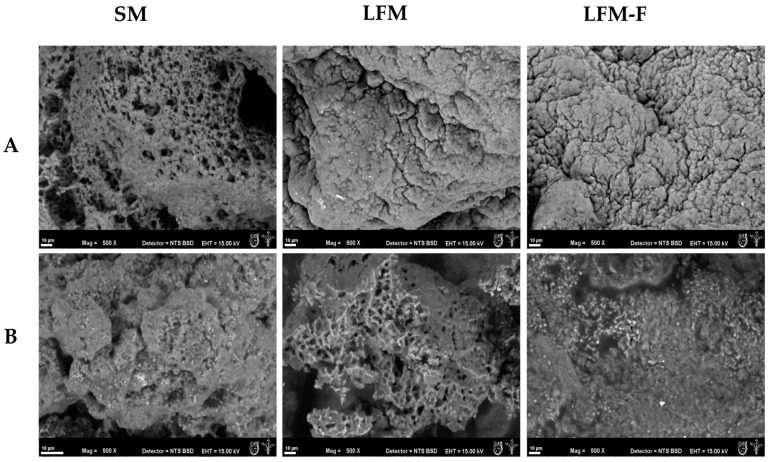
Micrographs by ESEM of: (**A**) surface section, (**B**) cross-section of milk kefir grains, where: SM: low-fat whole milk, LFM: lactose-free milk and LFM-F: lactose-free milk fortified with fructans. The scale bar indicates 10 µm.

**Figure 5 microorganisms-11-01570-f005:**
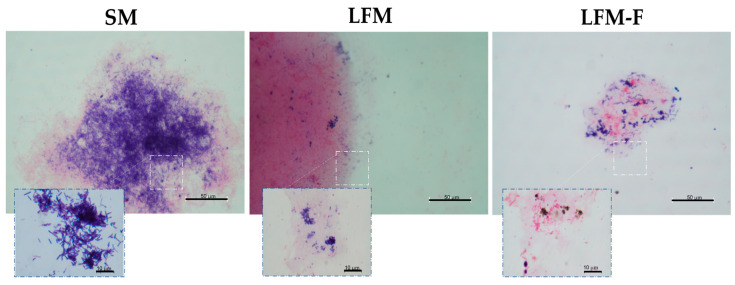
Micrographs GRAM via optical microscopy, where: SM: low-fat milk, LFM: lactose-free milk and LFM-F: lactose-free milk fortified with fructans. The scale bar indicates 50 µm, 20× and 10 µm, 100× in the zoom box.

**Table 1 microorganisms-11-01570-t001:** Chemical composition of kefir grain biomass.

Culture Medium	Moisture (%)	Fat (%)	Ashes (%)	Carbohydrates (%)	Protein (%)
SM	82.5 ± 0.90 ^a^	1.48 ± 0.39 ^c^	3.42 ± 0.31 ^b^	8.08 ± 0.70 ^b^	4.27 ± 0.71 ^b^
LFM	79.34 ± 1.16 ^b^	2.33 ± 0.33 ^b^	2.43 ± 0.12 ^c^	11.92 ± 0.91 ^a^	3.93 ± 0.16 ^b^
LFM-F	76.50 ± 2.10 ^b^	3.76 ± 0.52 ^a^	5.57 ± 0.21 ^a^	7.05 ± 0.81 ^b^	7.12 ± 0.36 ^a^

Values indicate the means ± SD, *n* = 3. Different letters represent statistically significant differences (*p* < 0.05). using an ANOVA followed by Tukey’s test. SM: Low-fat whole milk; LFM: Lactose-Free milk; LFM-F: Lactose-Free milk with Fructans.

**Table 2 microorganisms-11-01570-t002:** Textural features extracted from grayscale images of milk kefir grains surface section via GLCM algorithm.

Culture Medium	Contrast	Correlation	IDM	Entropy
SM	233.0488 ± 6.1819 ^a^	0.0004 ± 0.0001 ^a^	0.2953 ± 0.0167 ^a^	7.7130 ± 0.1534 ^a^
LFM	257.4760 ± 13.7788 ^a,b^	0.0004 ± 0.0000 ^a^	0.2490 ± 0.0147 ^b^	7.9323 ± 0.1106 ^a,b^
LFM-F	291.2643 ± 40.0676 ^b^	0.0003 ± 0.0000 ^a^	0.2230 ± 0.0216 ^b^	8.1703 ± 0.2742 ^b^

ANOVA test, average ± SD; *n* = 3. Values with different letters in the same column showed significant differences (*p*
< 0.05), where: IDM, Inverse Difference Moment; GLCM: Gray-Level Co-Occurrence Matrix; SM: Low-fat whole milk; LFM: Lactose-Free milk; LFM-F: Lactose-Free Milk with Fructans.

**Table 3 microorganisms-11-01570-t003:** Textural features extracted from grayscale images of milk kefir grain cross-section using GLCM algorithm.

Culture Medium	Contrast	Correlation	IDM	Entropy
SM	203.7258 ± 15.2888 ^a^	0.0008 ± 0.0002 ^a^	0.3504 ± 0.0244 ^a^	6.8348 ± 0.2234 ^a^
LFM	223.7620 ± 2.4241 ^b^	0.0005 ± 0.0000 ^b^	0.3070 ± 0.0140 ^b^	7.5570 ± 0.1208 ^a^
LFM-F	244.5800 ± 6.8698 ^b^	0.0005 ± 0.0001 ^b^	0.2423 ± 0.0149 ^b^	7.7743 ± 0.1148 ^a^

ANOVA test, average ± SD; *n* = 3. Values with different letters in the same column showed significant differences (*p*
< 0.05), where: IDM, Inverse Difference Moment; GLCM: Gray-Level Co-Occurrence Matrix; SM: Low-fat whole milk; LFM: Lactose-Free milk; LFM-F: Lactose-Free Milk with Fructans.

## Data Availability

Not applicable.

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
