# Peer review of "Effect of Agave Fructans on Changes in Chemistry, Morphology and Composition in the Biomass Growth of Milk Kefir Grains"

_microorganisms, 2023, doi:10.3390/microorganisms11061570_

Round 1

Reviewer 1 Report

In this manuscript the effect of lactose free milk, temperature of starter inoculum and the addition of substrate was evaluated in different proportions. Besides, the method of response surface analysis was performed to determine the optimum biomass production conditions was evaluated. However, I can send some observations to improve the manuscript:

Results and discussions:

Pag 4, line 172. Remove parentheses.

Figure 1. I recommend repainting in black axis "X" and "Y".

Figure 2. In English (temperature).

Pag 8, line 329. Check that all scientific names are in italics.

References:

Review in detail the format of the references, you have different types of letters, italics are missing in journal names, etc.

In general, it is a good document, I congratulate the authors for their work, it is well discussed, I have no further observations on the manuscript.

I have no comments about it.

Reviewer 2 Report

The manuscript is about the effect of agave fructans on milk kefir grains.

The study is well set up, but the description is very confused written. For the beginning of the abstract agave fructans are not even mentioned.

In the introduction is a well-described problem, but the basic is not mentioned, the difference between milk kefir grains and water kefir grains, and the two terms are discussed further in the manuscript.

In the materials and method is not explained why low-fat whole milk and lactose-free milk were used. It was mentioned only in the conclusions. Lactose-free milk with fructans is not mentioned. It is not mentioned in this section that SK was used as the control. It is first mentioned in line 320. Also, reference No 20 is not mentioned in the text. Additionally, tree repetitions in not 23 (Line 91).

When a number of references are listed, it should be written in one bracket, [13,22], instead [13][22] (line 97). Please correct that through the text. 

Almost the whole first part of the results and discussion explained the condition of growth, and the same things are repeated all over again. It should be summarised. 

 Figure 1 and Table 1 consist the same results. The Figure is enough. Some numerical results can be mentioned in the text.

The authors mentioned that some other kinds of carbohydrates were added to the milk, but that results are not discussed. Also, the term “substrate” is used for milk and for added carbohydrates. It should be distinguished.

In line 327 is mentioned glycerol in the medium. It is not mentioned in the materials and methods.

In section 3.6 are a lot of abbreviations (GLCM, IDM, SEM) that should be defined in the text. 

What kind of statistic was used should be described in the material and methods section, not at the bottom of the tables.

Reviewer 3 Report

The authors were trying to study the factors that influence the increase in biomass in milk kefir grains. Specifically, they only focused on the impacts of carbon sources on biomass in milk kefir grains. Based on this objective, they used a full factorial design with response surface methodology analysis. They found that a 2% inoculum was the most effective and the optimum fermentation temperature was 25°C with the addition of 6% w/w fructans. 

In general, the design is clear and reasonable to explore the impacts of carbon source on biomass. But the results are not surprising and are not at the publishing level. 

The experimental design missed a lot of details, such as the incubation process parameter settings, the control of other parameters, the type of bioreactor or incubation equipment, etc.

Therefore, I would not recommend publish on this journal. 

Grammar mistakes were found in several places.

Reviewer 4 Report

Although this study on Kefir is complete and it also has the necessary results and conclusions, I do not think it is sufficiently innovative cause the optimization, the property analyses are normal, uncomplicated and predictable. I regret that It is not suitable for publication in this journal.

Round 2

Reviewer 2 Report

It has been seen that the authors improved the manuscript, but still, some things are not clear.

Specific comments:

In the section materials and methods are still unclear: SM milk was the control, and LFM is a substrate in which authors added carbohydrates? Or you tested for the beginning growth of substrate in each milk without any additives? So, all results are explained in that way, or what? Please be more specific. Because in line 246 you have referred to a reference in which is shown that milk with some carbohydrates has better results in fermentation than milk without carbohydrates. Therefore, please describe methods more clearly.

Line 138: what means “batch fermentation”?

Lines 239-240: abbreviations are enough, no explanations are needed

Line 412: please replace “substrate” with some other word, maybe “carbohydrates” (or some other synonym) to be more clearly

Section 3: it is not clear how you interpret results. Carbohydrates were added in FLM or not? That is why is important to describe the materials as detailed as possible.

English very difficult to understand/incomprehensible

Reviewer 3 Report

Please reference my previous comments.

Reviewer 4 Report

The revised manuscript can be accepted after improving the quality of figures, especially figure 1 and 2.

Minor editing of English language required
